# Global Emissions of Perfluorocyclobutane (PFC-318, $c$-C$_4$F$_8$) Resulting from the Use of Hydrochlorofluorocarbon-22 (HCFC-22) Feedstock to Produce Polytetrafluoroethylene (PTFE) and related Fluorochemicals

Jens Mühle[1*], Lambert J. M. Kuijpers[2], Kieran M. Stanley[3], Matthew Rigby[4], Luke M. Western[4], Jooil Kim[1], Sunyoung Park[5], Christina M. Harth[1], Paul B. Krummel[6], Paul J. Fraser[6], Simon O'Doherty[3], Peter K. Salameh[1], Roland Schmidt[1], Dickon Young[3], Ronald G. Prinn[7], Ray H. J. Wang[8], and Ray F. Weiss[1]

[1]Scripps Institution of Oceanography, University of California San Diego, La Jolla, CA, 92093, USA
[2]A/gent Consultancy BV, 5911BA Venlo, The Netherlands
[3]Institute for Atmospheric and Environmental Sciences, Goethe University Frankfurt, Frankfurt, 60438, Germany
[4]School of Chemistry, University of Bristol, Bristol, BS8 1TS, UK
[5]Department of Oceanography, Kyungpook National University, Daegu, 41566, Republic of Korea
[6]Climate Science Centre, CSIRO Oceans and Atmosphere, Aspendale, Victoria, 3195, Australia
[7]Center for Global Change Science, Massachusetts Institute of Technology, Cambridge, MA, 02139, USA
[8]School of Earth and Atmospheric Sciences, Georgia Institute of Technology, Atlanta, GA, 30332, USA

Correspondence to: Jens Muhle (jmuhle@ucsd.edu)

## Abstract

Emissions of the potent greenhouse gas perfluorocyclobutane ($c$-C$_4$F$_8$, PFC-318, octafluorocyclobutane) into the global atmosphere inferred from atmospheric measurements have been increasing sharply since the early 2000s. We find that these inferred emissions are highly correlated with the production of hydrochlorofluorocarbon-22 (HCFC-22, CHClF$_2$) for feedstock (FS) uses, because almost all HCFC-22 FS is pyrolyzed to produce (poly)tetrafluoroethylene ((P)TFE) and hexafluoropropylene (HFP), a process in which $c$-C$_4$F$_8$ is a known by-product, causing a significant fraction of global $c$-C$_4$F$_8$ emissions. We find a global emission factor of ~0.003 kg $c$-C$_4$F$_8$ per kg of HCFC-22 FS pyrolyzed. Mitigation of these $c$-C$_4$F$_8$ emissions, e.g., through process optimization, abatement, or different manufacturing processes, such as refined methods of electrochemical fluorination and waste recycling, could reduce the climate impact of this industry. While it has been shown that $c$-C$_4$F$_8$ emissions from developing countries dominate global emissions, more atmospheric measurements and/or detailed process statistics are needed to quantify $c$-C$_4$F$_8$ emissions at country to facility levels.

## 1 Introduction

Perfluorocyclobutane ($c$-C$_4$F$_8$, PFC-318, octafluorocyclobutane, CAS 115-25-3) is a potent greenhouse gas (GHG) with a global warming potential of 10,200 on a 100-year timescale (GWP$_{100}$) based on a lifetime estimate of 3200 years (Forster et

al., 2021). Mühle et al. (2019) reported that global atmospheric emissions of $c$-$C_4F_8$ began in the late-1960s, reaching a plateau of ~1.2 Gg yr$^{-1}$ during late-1970s to the late-1980s, followed by a decline to a plateau of ~0.8 Gg yr$^{-1}$ during the early-1990s to early-2000s, and then increased sharply reaching ~2.2 Gg yr$^{-1}$ in 2017. Emissions of $c$-$C_4F_8$ from developed countries are reported under the United Nations Framework Convention on Climate Change (UNFCCC). However, these reports from developed countries account only for a small fraction of global emissions of $c$-$C_4F_8$ inferred from atmospheric measurements (Mühle et al., 2019), similar to the emissions gaps observed for other synthetic GHGs (e.g., Montzka et al., 2018; Mühle et al., 2010; Stanley et al., 2020). This emissions gap results partly from emissions in developing countries, which do not have to be reported to the UNFCCC and are therefore missing, and/or from uncertainties in emissions reported by developed countries. To understand the sources of recent global $c$-$C_4F_8$ emissions, Mühle et al. (2019) used Bayesian inversions of atmospheric $c$-$C_4F_8$ measurements made at sites of the Advanced Global Atmospheric Gases Experiment (AGAGE, Prinn et al., 2018) in East Asia and Europe and from an aircraft campaign over India. For 2016, these limited regional measurements allowed Mühle et al. (2019) to allocate ~56% of global $c$-$C_4F_8$ emissions to specific regions with significant emissions from Eastern China (~32%), Russia (~12%), and India (~7%). Spatial patterns of these regional $c$-$C_4F_8$ emissions were roughly consistent with locations of facilities that produce polytetrafluoroethylene (PTFE, a polymer widely used for its non-stick and water repellent properties, chemical, thermal, light, and electrical resistance, high flexibility and low friction), related fluoropolymers and the necessary precursor monomers tetrafluoroethylene (TFE) and hexafluoropro-pylene (HFP), which are produced via the pyrolysis of hydrochlorofluorocarbon-22 (HCFC-22, $CHClF_2$). $c$-$C_4F_8$, essentially the dimer of TFE, is one of several by-products/intermediates of this process (Chinoy and Sunavala, 1987; Broyer et al., 1988; Gangal and Brothers, 2015; Harnisch, 1999; Ebnesajjad, 2015). Process control and optimization to reduce the formation of $c$-$C_4F_8$ and other by-products are complex, and under unsuitable conditions $c$-$C_4F_8$ by-production could be as high as 14% (Ebnesajjad, 2015). On the other hand, Murphy et al. (1997) demonstrated that co-feeding several percent of $c$-$C_4F_8$ to the HCFC-22 feed could reduce additional $c$-$C_4F_8$ formation to less than 0.5% of the combined TFE and HFP yield, thus increasing combined TFE and HFP yield to more than 96%. But they also stated that perfect process control may be impractical. In 2018, one of China's largest TFE producers confirmed $c$-$C_4F_8$ by-product formation (Mühle et al., 2019). Unless $c$-$C_4F_8$ is recovered or recycled, excess $c$-$C_4F_8$ may therefore be emitted to the atmosphere, consistent with the observations. Historically, similar $c$-$C_4F_8$ by-product venting occurred in the US and Europe (Mühle et al., 2019), unnecessarily increasing the carbon footprint of this industry. Note that Ebnesajjad (2015) and e.g., Mierdel et al. (2019) discuss research into the use of refined methods of electrochemical fluorination (ECF) and waste recycling which may offer significantly reduced by-product formation rates in addition to energy savings and overall waste reduction.

Closely related to $c$-$C_4F_8$ (as a by-product of HCFC-22 pyrolysis) is hydrofluorocarbon-23 (HFC-23, $CHF_3$), a strong GHG as well, which has long been known to be a by-product of the total (FS and non-FS) production of HCFC-22 from chloroform ($CHCl_3$), that is also often vented to the atmosphere, despite the existence of technical solutions, regulations, and financial incentives (e.g., Stanley et al., 2020).

Here we show that global emissions of $c$-$C_4F_8$ since 2002 are highly correlated with the amount of HCFC-22 produced for feedstock (FS) uses, because almost all this FS HCFC-22 is pyrolyzed to produce TFE/HFP, a process with $c$-$C_4F_8$ as a known by-product. This supports the hypothesis that recent global $c$-$C_4F_8$ emissions are dominated by $c$-$C_4F_8$ by-product emissions from the production of TFE/HFP, PTFE and related fluoropolymers and fluorochemicals.

## 2 Methods

### 2.1 Atmospheric observations of $c$-$C_4F_8$ and inverse modeling of global emissions

We have extended the 1970-2017 AGAGE in situ $c$-$C_4F_8$ atmospheric measurement record used by Mühle et al. (2019) and produced updated global emissions through 2020. For this we used measurements of $c$-$C_4F_8$ by "Medusa" gas chromatographic systems with quadrupole mass selective detection (GC/MSD) (Arnold et al., 2012; Miller et al., 2008) from five AGAGE stations: Mace Head, Ireland (MHD, 53.3°N, 9.9°W); Trinidad Head, USA (THD, California, 41.0°N,
124.1°W); Ragged Point, Barbados (RPB, 13.2°N, 59.4°W); Cape Matatula, American Samoa (SMO, 14.2°S, 170.6°W); Cape Grim, Australia (CGO, Tasmania, 40.7°S, 144.7°E). Ambient air and reference gas measurements are alternated resulting in up to 12 fully calibrated samples per day (Prinn et al., 2018). Reference gases are supplied by the Scripps Institution of Oceanography (SIO) and all $c$-$C_4F_8$ data are reported on the SIO-14 calibration scale in parts-per-trillion (ppt) dry-air mole fractions. Daily reference gas measurement precisions are ~0.01–0.02 ppt (~1–2%); for more details see Mühle
et al. (2019).

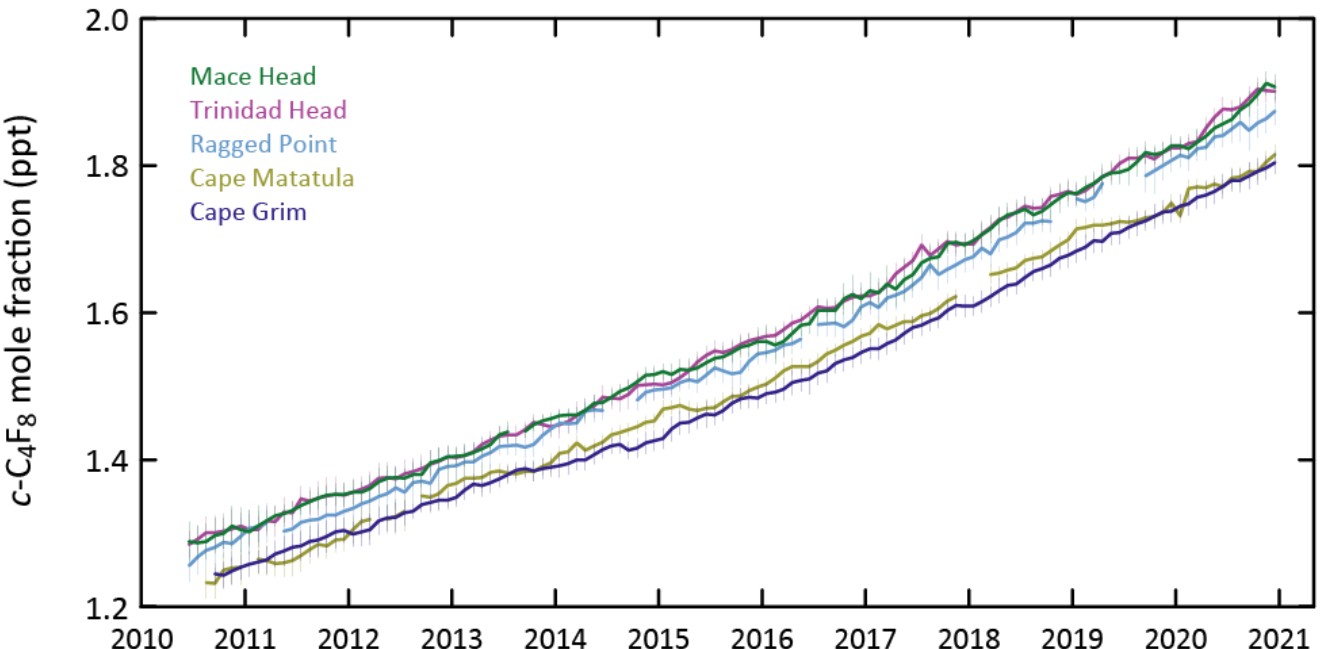

**Figure 1** Pollution free monthly mean mole fraction calculated from in situ $c$-$C_4F_8$ measurements at five AGAGE sites using the AGAGE statistical method (Cunnold et al., 2002) (https://agage.mit.edu/data/agage-data).

In situ data were filtered with the AGAGE statistical method to remove pollution events (Cunnold et al., 2002). For these baseline stations more than 99% of the data were retained, reflecting minor regional $c$-$C_4F_8$ emissions near these stations. In contrast, in East Asia strong and frequent pollution events were observed and corresponding strong emissions were inferred, as detailed in Mühle et al. (2019). Fig. 1 shows the continued increase of pollution free monthly mean $c$-$C_4F_8$ mole fractions in the global atmosphere since the start of in-situ measurements. Atmospheric abundances before in-situ measurements were

reconstructed based on measurements of samples of the Cape Grim Air Archive (CGAA) for the extratropical Southern Hemisphere and archived air samples from various sources for the extratropical Northern Hemisphere (not shown) as detailed in Mühle et al. (2019). The data were then used in conjunction with the AGAGE 12-box two-dimensional model (Rigby et al., 2013) and a Bayesian inverse method to update global emissions (Table 1 and Fig. 2). The model describes the transport and loss of trace gases in the global atmosphere and calculates mole fractions in each model box with latitudinal

divisions at 30˚S, 0˚ and 30˚N and pressure divisions at 500 and 200 hPa. Model transport parameters are varied seasonally but repeated annually. In the model the $c$-$C_4F_8$ lifetime is set to infinity. Details of this inversion are given in Rigby et al. (2014) and Mühle et al. (2019).

| | $c$-$C_4F_8$ emissions (Gg yr$^{-1}$, 1 σ) global | HCFC-22 feedstock (FS) production (Gg yr$^{-1}$, ktonnes yr$^{-1}$) | | | |
|---|---|---|---|---|---|
| | | non-A5 (developed) countries[a] | A5 (developing) countries[a] | A5 China only[b] | Global (non-A5 + A5)[a] |
| 1990 | 0.93 (0.76 - 1.11) | 23.3 | 0.0 | | 23.3 |
| 1991 | 0.87 (0.71 - 1.03) | 69.2 | 7.0 | | 76.2 |
| 1992 | 0.80 (0.65 - 0.97) | 49.9 | 11.2 | | 61.1 |
| 1993 | 0.76 (0.59 - 0.93) | 40.1 | 10.5 | | 50.6 |
| 1994 | 0.74 (0.57 - 0.89) | 85.2 | 12.1 | | 97.3 |
| 1995 | 0.74 (0.57 - 0.90) | 61.2 | 21.7 | | 82.9 |
| 1996 | 0.76 (0.61 - 0.91) | 129.8 | 21.7 | | 151.5 |
| 1997 | 0.77 (0.63 - 0.89) | 147.7 | 18.8 | | 166.5 |
| 1998 | 0.76 (0.61 - 0.90) | 154.7 | 1.1[c] | | 155.7 |

| 1999 | 0.75 (0.60 - 0.89) | 158.5 | 16.2 | | 174.7 |
|------|--------------------|-------|------|-----|-------|
| 2000 | 0.74 (0.61 - 0.89) | 135.2 | 0.1[c] | | 135.3 |
| 2001 | 0.74 (0.61 - 0.93) | 152.4 | 0.3[c] | | 152.7 |
| 2002 | 0.77 (0.63 - 0.97) | 163.1 | 34.2 | | 197.3 |
| 2003 | 0.82 (0.66 - 0.97) | 171.3 | 43.1 | | 214.4 |
| 2004 | 0.89 (0.75 - 1.06) | 203.1 | 59.8 | | 262.9 |
| 2005 | 0.96 (0.83 - 1.14) | 192.8 | 78.3 | | 271.1 |
| 2006 | 1.03 (0.91 - 1.20) | 193.1 | 92.1 | | 285.2 |
| 2007 | 1.09 (0.95 - 1.23) | 186.1 | 110.5 | | 296.6 |
| 2008 | 1.17 (1.03 - 1.30) | 174.2 | 194.3 | 166.1 | 368.5 |
| 2009 | 1.28 (1.13 - 1.43) | 121.0 | 186.6 | 171.9 | 307.6 |
| 2010 | 1.43 (1.30 - 1.58) | 165.2 | 244.9 | 214.7 | 410.2 |
| 2011 | 1.56 (1.46 - 1.71) | 191.1 | 291.6 | 242.2 | 482.7 |
| 2012 | 1.65 (1.54 - 1.77) | 180.1 | 302.2 | 262.2 | 482.4 |
| 2013 | 1.69 (1.58 - 1.82) | 161.7 | 345.3 | 308.0 | 506.9 |
| 2014 | 1.77 (1.68 - 1.92) | 179.2 | 357.6 | 302.9 | 536.8 |
| 2015 | 1.89 (1.79 - 2.04) | 201.9 | 316.0 | 270.7 | 517.9 |
| 2016 | 2.09 (1.97 - 2.24) | 193.4 | 365.9 | 290.3 | 559.4 |
| 2017 | 2.26 (2.13 - 2.39) | 207.1 | 438.9 | 372.3 | 646.0 |
| 2018 | 2.28 (2.16 - 2.43) | 208.5 | 484.5 | 339.7 | 693.0 |
| 2019 | 2.26 (2.11 - 2.40) | 200.1 | 512.6 | | 712.7 |
| 2020 | 2.32 (2.16 - 2.48) | | | | |

**Table 1** Global $c$-$C_4F_8$ emissions determined from AGAGE atmospheric measurements and hydrochlorofluorocarbon-22 (HCFC-22) feedstock (FS) production from United Nations Environment Programme (UNEP) and Technology and Economic Assessment Panel (TEAP) reports. Most of HCFC-22 feedstock (FS) production in developing (A5) countries occurs in China.

[a]UNEP (2021). [b]See Table 4-1 TEAP (2020). HCFC-22 FS production data for China before 2008 is not publicly available. [c]China accounted for >90% of A5 HCFC-22 production during 1991 to 2007, but did not report for 1998, 2000, and 2001 to UNEP, leading to the low A5 values for these years.

## 2.2 HCFC-22 feedstock (FS) production data

To investigate whether the chemical relationship between HCFC-22 pyrolysis and $c$-$C_4F_8$ by-product (as discussed in the Introduction) results in a correlation between HCFC-22 feedstock (FS) production and $c$-$C_4F_8$ emissions, we compiled

HCFC-22 FS production statistics (Table 1 and Fig. 2). While production of HCFC-22 for such presumed non-emissive FS uses are not regulated by the Montreal Protocol on Substances that Deplete the Ozone Layer (MP), various types of data, including FS production, are reported by all countries (parties) to the United Nations Environment Programme (UNEP) under Article 7 of the MP. Specifically, HCFC-22 FS production data for MP Article 5 (A5, developing) countries and non-Article 5 (non-A5, developed) countries were used here (UNEP, 2021). Additionally, HCFC-22 FS production data for China were taken from Table 4-1 in the TEAP (2020) report for 2008 to 2018; this report contains data used for the determination of the funding requirement for the Multilateral Fund (MLF) for the implementation of the MP. It also lists totals for A5 countries which show small inconsistencies with the UNEP (2021) data, probably due to recent updates. Data for the last year or two are often adjusted in the next report. Table 1 shows that Chinese HCFC-22 FS production from 2008 to 2018 accounted for $(84 \pm 6)\%$ of the A5 (developing countries) HCFC-22 FS production $((86 \pm 3)\%$ if the last year, 2018, is excluded), i.e., most of the HCFC-22 feedstock (FS) production in developing (A5) countries occurs in China.

Note, that we do not discuss HCFC-22 non-FS production statistics, i.e., HCFC-22 produced for emissive uses (e.g., refrigeration and foam blowing). While critical for understanding HCFC-22 emissions and HCFC-22 atmospheric burden, amounts of HCFC-22 produced for non-FS uses are not relevant for $c$-$C_4F_8$ emissions. We also do not discuss total HCFC-22 (non-FS plus FS) production. While critical for understanding HFC-23 by-product emissions (from total HCFC-22 production) and HFC-23 atmospheric burden, they are not directly relevant for $c$-$C_4F_8$ emission studies. Only HCFC-22 that is produced for FS uses and pyrolyzed to TFE/HFP with $c$-$C_4F_8$ by-product is relevant for $c$-$C_4F_8$ emissions and $c$-$C_4F_8$ atmospheric burden. It is worth noting though that the global HCFC-22 market is complex. For example, the decrease in HCFC-22 FS production in 2009 (developed countries and total global) was preceded by a large increase in HCFC-22 FS production in developing countries in 2008 (Table 1 and Fig. 2). This was a result of increased Chinese HCFC-22 production for demand-based FS uses, most notably PTFE, which may have displaced exports into China. Outside of China, there was also a shortage of hydrogen fluoride, needed to produce HCFC-22 and almost all other fluorocarbons (David Sherry, personal communication, 2022). It is also possible that some of the HCFC-22 FS produced at the year-end was used (pyrolyzed) in the next year.

**3 Results and Discussion**

Our updated global inversion results show that $c$-$C_4F_8$ emissions were relatively stable at ~0.8 Gg yr$^{-1}$ in the early-1990s to early-2000s. However, in 2002, $c$-$C_4F_8$ emission growth resumed, reaching levels not seen before, with a relatively steady increase to 2.26 Gg yr$^{-1}$ in 2017 (Table 1 and Fig. 2, black diamonds; these emissions are very similar those in Mühle et al. (2019), which were based on a mostly identical, albeit shorter duration, AGAGE data set and inverse method). Here, we find a stabilization at this emission level from 2017 to 2019, followed by a possible resumed increase in emission growth to 2.32 Gg yr$^{-1}$ (24 million metric tons of $CO_2$-equivalents yr$^{-1}$) in 2020 (however, differences between the 2017-2020 emissions are not statistically significant). In comparison, global HCFC-22 production for feedstock (FS) uses has increased relatively

steadily since the early 1990s, initially driven by FS production in developed (non-A5) countries (Fig. 2, red circles). This growth in developed (non-A5) countries slowed down in the early-2000s and HCFC-22 FS production in developed countries has been relatively stable since then. The global growth in HCFC-22 FS production since 2002 has been driven by the increase in production in developing (A5) countries (Fig. 2, blue squares), dominated by China (Fig. 2, open orange squares). Coincidentally or not, this is the time frame of a steady increase of inferred global $c$-$C_4F_8$ emissions.

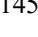

**Figure 2** HCFC-22 feedstock (FS) production (Gg yr$^{-1}$). Global HCFC-22 FS production (green triangles) is the sum of HCFC-22 FS production in non-A5 (developed, red circles) and A5 (developing, blue squares) countries. Since about 2002, the increasing trend of global HCFC-22 FS production is dominated by growth in A5 countries, particularly China (orange open squares), while HCFC-22 FS
production in non-A5 countries has been relatively stable.

We find a strong correlation between global HCFC-22 FS production and inferred global $c$-$C_4F_8$ emissions ($R^2$ = 0.97, p < 0.01) (Fig. 3, green triangles and fit, 2002-2019). While HCFC-22 FS production itself does not lead to $c$-$C_4F_8$ by-production

and emissions, it is estimated that almost all (David Sherry, Andy Lindley, personal communications, 2022) of global
HCFC-22 FS production is used to produce TFE and HFP, to in turn produce PTFE and related fluoropolymers and
fluorochemicals, which causes the observed strong correlation of HCFC-22 FS production with $c$-$C_4F_8$ emissions. This
would probably not be the case if a significant fraction of HCFC-22 FS production were used for other processes without
$c$-$C_4F_8$ by-production and emissions. Note that the HCFC-22 to TFE route (with $c$-$C_4F_8$ by-product) can also be used to
produce HCFC-225 isomers and hydrofluoroolefin HFO-1234yf ($CF_3$-$CF$=$CH_2$) (Sherry et al., 2019), with HFO-1234yf
being the preferred replacement for HFC-134a ($CF_3$-$CFH_2$) in mobile air conditioning (MAC).

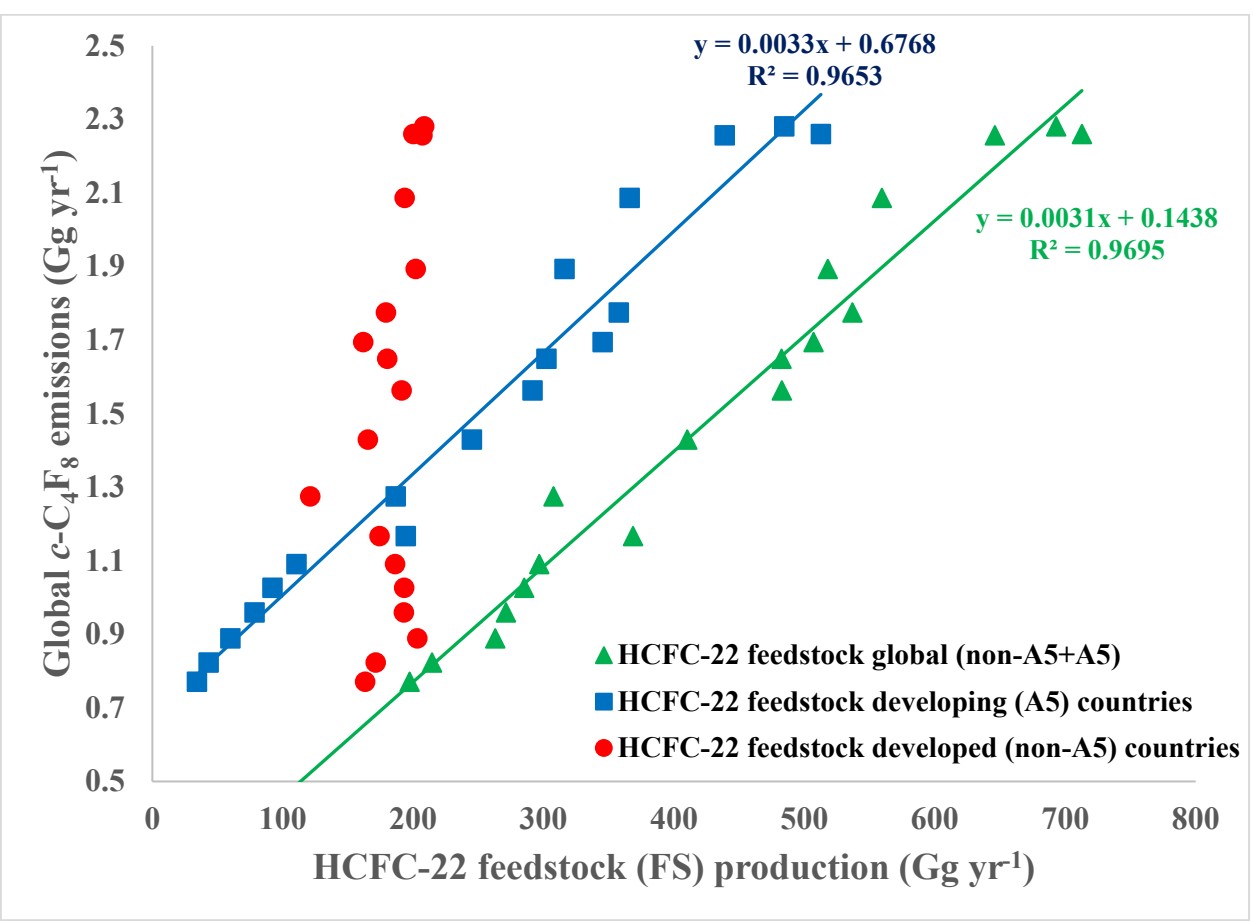

**Figure 3** The relationship between A5 (developing, blue squares), non-A5 (developed, red circles) countries and total global HCFC-22
feedstock (FS, green triangles) production and global $c$-$C_4F_8$ (PFC-318) emissions (2002-2019).

Current estimates are that perhaps 3% of HCFC-22 FS produced is used in reactions other than the TFE/HFP route (David Sherry, Andy Lindley, personal communications, 2022) that is without $c$-$C_4F_8$ by-product; products include sulfentrazone herbicide, pantoprazole (acid reflux) pharmaceutical, isoflurane and desflurane anesthetics, as well as high-purity HFC-23 for refrigeration use and as feedstock to manufacture iodotrifluoromethane, halon-1301 and from this, fipronil pesticide, mefloquine (antimalarial) and DPP-IV inhibitor (antidiabetic) pharmaceuticals (TEAP, 2021).

The observed post-2001 correlation between $c$-$C_4F_8$ emissions and HCFC-22 FS use supports our hypothesis that current global emissions of $c$-$C_4F_8$ are dominated by HCFC-22 FS use to produce TFE/HFP and related products. The correlation indicates an emission factor (EF) of (0.0031 ± 0.0001) kg $c$-$C_4F_8$ emitted per kg of HCFC-22 produced for FS use (to produce TFE/HFP) with an intercept of 0.14 Gg yr$^{-1}$ $c$-$C_4F_8$, presumably reflecting $c$-$C_4F_8$ emissions from other sources, such as semiconductor (SC), photovoltaic (PV), liquid crystal display (LCD), and micro-electromechanical system (MEMS) production. The annual reports of the World Semiconductor Council (WSC) ([http://www.semiconductorcouncil.org/public-documents/joint-statements-from-prior-wsc-meetings/](http://www.semiconductorcouncil.org/public-documents/joint-statements-from-prior-wsc-meetings/)) contain estimates of $c$-$C_4F_8$ emissions from SC production in China, Taiwan, Europe, Japan, South Korea, and the United States. They range from ~0.05 Gg yr$^{-1}$ in 2012-2014 to ~0.11 Gg yr$^{-1}$ in 2018-2019, somewhat smaller than the 0.14 Gg yr$^{-1}$ $c$-$C_4F_8$ intercept. We also updated the global $c$-$C_4F_8$ bottom-up inventory from Mühle et al. (2019) using the 2021 National Inventory Submissions to UNFCCC ([https://unfccc.int/ghg-inventories-annex-i-parties/2021](https://unfccc.int/ghg-inventories-annex-i-parties/2021)) and then augmented this with their top-down emission estimates for Western Japan, South Korea, North Korea, and Taiwan (but not China). The resulting emission estimates are ~0.09 Gg yr$^{-1}$ in 2012-2019 and include top-down $c$-$C_4F_8$ emission estimates from all processes such as SC, PV, LCD, and MEMS production in these four countries, but also from any HCFC-22 FS pyrolysis in these countries, most notably in Japan. We did not include U.S. EPA emission estimates of ~0.06 Gg yr$^{-1}$ $c$-$C_4F_8$ from U.S. fluorinated gas producers ([https://www.epa.gov/ghgreporting/data-sets](https://www.epa.gov/ghgreporting/data-sets)) in this updated estimate, as most of these $c$-$C_4F_8$ emissions stem from facilities that pyrolyze HCFC-22 (Deborah Ottinger, personal communication, 2022). Overall, the data support our conclusion that currently $c$-$C_4F_8$ emissions from sources other than HCFC-22 FS use (to produce TFE/HFP) are small, perhaps ~0.1-0.14 Gg yr$^{-1}$.

Note that a fit of HCFC-22 FS production in developing (A5) countries and global $c$-$C_4F_8$ emissions results in a similar EF (slope) of (0.0033 ± 0.0002) kg/kg ($R^2$ = 0.97, $p < 0.01$, blue squares and fit, 2002-2019). The reason is that HCFC-22 FS use in developed (non-A5, see Fig. 2, red circles) countries has been essentially stable since the early 2000s (Fig. 3, red circles, 2002-2019) causing a change in the offset rather than in the slope (EF). We therefore cannot determine whether current $c$-$C_4F_8$ emission factors from HCFC-22 FS use in developing (A5) and developed (non-A5) countries are similar or not. Atmospheric measurements covering individual countries and facilities are needed to determine this.

The global EF of ~0.003 kg/kg or ~0.3% (by weight) of $c$-$C_4F_8$ emitted per HCFC-22 FS used are similar to the optimal production conditions explored by Murphy et al. (1997) of less than 0.5% $c$-$C_4F_8$ by-product of the combined TFE and HFP yield (excluding other by-products). Historic $c$-$C_4F_8$ EFs were probably much higher, particularly during the early decades of PTFE production (1950-1990) when process controls or abatement were likely not in place. From the 1980s onwards, it is likely that EFs steadily improved with the advent of UNFCCC emission reporting requirements in the 1990s, concerns about

the environment, climate change and product stewardship, abatement, and perhaps collection of $c$-$C_4F_8$ for use in the semiconductor industry, where it can be easily abated (Mühle et al., 2019, David Sherry, personal communication, 2022). We can investigate the EF for the period from 1996 to 2001, before the start of any significant production of HCFC-22 for

FS uses in developing (A5) countries, as $c$-$C_4F_8$ emissions and developed (non-A5) HCFC-22 FS production were both relatively stable (Fig. 2). Assuming that all of the HCFC-22 produced for FS uses in developed (non-A5) countries was pyrolyzed to TFE/HFP with $c$-$C_4F_8$ by-product emissions and that other sources of $c$-$C_4F_8$ were small, an EF of 0.0052 ± 0.0004 kg/kg could be calculated, which is larger than the global EF in recent years, suggesting that EF reductions were still progressing.

Lastly, using the HCFC-22 FS production data for China (Table 1) and the top-down $c$-$C_4F_8$ emission estimates from Mühle et al. (2019) we can also investigate emission factors for China. This is of interest as (84 ± 6)% of HCFC-22 FS production in developing (A5) countries occurred in China (2008-2018). A caveat is that the underlying atmospheric measurements were mostly sensitive to emissions in Eastern China, which means that emissions from several production complexes in other parts of China (see the Supplement and Fig. 7 in Mühle et al., 2019) with likely $c$-$C_4F_8$ emissions are probably missing.

Still, dividing the $c$-$C_4F_8$ emissions for eastern China of 0.67 ± 0.13 (~32% of global emissions, Mühle et al., 2019) for 2016/2017 by the HCFC-22 FS production reported by China for these years (Table 1), results in an EF of 0.0021 ± 0.0003 kg/kg. This is lower than the EF determined for the total global (or all developing (A5) countries) in recent years, which seems unlikely, since the increase in global (and A5 country) HCFC-22 FS production is driven by increases in China (Table 1, Fig. 2). Most probably, total Chinese $c$-$C_4F_8$ emissions are larger than those determined for eastern China. More

atmospheric measurements covering other parts of China are needed to investigate this.

## 4 Summary and Conclusions

Emissions of $c$-$C_4F_8$ (PFC-318, perfluorocyclobutane) into the global atmosphere have steadily increased since 2002 from 0.77 Gg yr$^{-1}$ to 2.32 Gg yr$^{-1}$ in 2020 (24 million metric tons of $CO_2$-equivalents yr$^{-1}$). We find that the chemical relationship between industrial scale HCFC-22 pyrolysis and $c$-$C_4F_8$ by-production leads to a tight correlation between global HCFC-22

feedstock (FS) production and global $c$-$C_4F_8$ emissions from 2002 to 2019. This correlation arises as almost all of the HCFC-22 FS production is used to produce TFE and HFP via HCFC-22 pyrolysis, with $c$-$C_4F_8$ as by-product. Emission factors are estimated to be ~0.003 kg $c$-$C_4F_8$ emitted per kg of HCFC-22 FS (to produce TFE and HFP) or ~0.3% (by weight). In 2018, one of the largest TFE producer in China confirmed $c$-$C_4F_8$ by-product formation, which, unless recovered or recycled, may lead to $c$-$C_4F_8$ emissions. Historically, similar $c$-$C_4F_8$ by-product venting occurred in the US and Europe and may still occur.

Based on the available atmospheric measurements we cannot determine whether current EFs in developed (non-A5) and developing (A5) countries are similar or dissimilar. Atmospheric measurements covering individual countries and facilities are needed to investigate this.

Closely related to emissions of $c$-$C_4F_8$ are emissions of hydrofluorocarbon-23 (HFC-23), also a strong GHG, which has long been a known by-product of the actual production of HCFC-22 from chloroform ($CHCl_3$). Emissions of HFC-23 contribute

unnecessarily to the carbon footprint of HCFC-22 industry despite technical solutions, regulations, and financial incentives (e.g., Stanley et al., 2020). Similarly, we have shown strong evidence that use of HCFC-22 feedstock for pyrolysis to TFE/HFP to produce fluoropolymers and related fluorochemicals likely causes most of the global $c$-$C_4F_8$ emissions. To reduce overall global GHG emissions of the HCFC-22/TFE/HFP/PTFE industry, further efforts to mitigate $c$-$C_4F_8$ and HFC-23 emissions should be considered, e.g., through process optimization, abatement, or different manufacturing processes such

as refined methods of electrochemical fluorination and waste recycling.

## Data and code availability

The data used in this work are available in the Supplement. Most up-to-date and quality-controlled AGAGE data are available at http://agage.mit.edu/data/agage-data (http://agage.eas.gatech.edu/data_archive/agage/gc-ms-medusa/complete/, http://agage.eas.gatech.edu/data_archive/agage/gc-ms-medusa/monthly/) and/or upon request. AGAGE data are also

regularly submitted to https://data.ess-dive.lbl.gov/data; at the time of writing, the most recent AGAGE data are available at https://data.ess-dive.lbl.gov/view/doi:10.15485/1841748. AGAGE 12-box model code can be made available upon request by contacting MR.

## Author contributions

Measurements and/or oversight for measurement collection were provided by JM, KMS, JK, SP, CMH, PBK, PJF, SOD, RS,

and DY. CMH provided and maintained the gravimetric SIO calibration scale for $c$-$C_4F_8$. RHJW processed the AGAGE data and produced pollution free monthly mean $c$-$C_4F_8$ abundances. MR and LMW performed model analysis. PKS wrote the GCWerks software to control the instruments, acquire the data, collect the data from all stations, and perform calculations necessary to provide calibrated end results. JM conceptualized the work, analysed the data, visualized the data, and wrote the manuscript with contributions from LJMK and all other co-authors. LJMK provided most valuable insight into industrial

processes and collected UNEP data. RGP and RFW were responsible for the overall management and the funding for this work.

## Competing interests

The authors declare that they have no conflict of interest.

**Acknowledgments**

Overall operation of the AGAGE network, including the measurements at Mace Head, Trinidad Head, Cape Matatula, Ragged Point, and Cape Grim were supported by National Aeronautics and Space Administration grants (nos. NNX16AC96G and NNX16AC97G to SIO and NNX16AC98G to MIT). Additional funding was provided by the Department for Business, Energy & Industrial Strategy (BEIS) Contract 1537/06/2018 (to the University of Bristol for Mace Head) and the National Oceanic and Atmospheric Administration (NOAA, contract 1305M319CNRMJ0028 to the 265 University of Bristol for Ragged Point). We thank the Commonwealth Scientific and Industrial Research Organisation (CSIRO, Australia) and the Bureau of Meteorology (Australia) for their ongoing long-term support and funding of the Cape Grim station and the Cape Grim science program. S. Park and operations of the Gosan station on Jeju Island, South Korea were supported by the National Research Foundation of Korea (NRF) grant funded by the Korean government (MSIT) (no. 2020R1A2C3003774). L. J. M. Kuijpers was supported by A/gent. M. Rigby and L.M. Western were supported by UK 270 Natural Environment Research Council grants NE/S004211/1, NE/V002996/1 and NE/N016548/1. We are indebted to the staff and scientists at AGAGE and other sites for their continuing contributions to produce high-quality measurements of atmospheric trace gases. We thank David Sherry (Nolan Sherry & Associates), Andy Lindley (UNEP Medical and Chemicals Technical Options Committee, MCTOC) and Deborah Ottinger (U.S. EPA), as well as four anonymous reviewers for their invaluable insights and excellent suggestions how to improve the manuscript.

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
