# Peer review of "Global Emissions of Perfluorocyclobutane (PFC-318, *c*-C4F8) Resulting from the Use of Hydrochlorofluorocarbon-22 (HCFC-22) Feedstock to Produce Polytetrafluoroethylene (PTFE) and related Fluorochemicals"

_Atmospheric Chemistry and Physics, 2021_

## Author Comment (AC1)

We thank the four reviewers for their overall positive reviews and very helpful suggestions. Below we repeat the comments, questions, and suggestions from each reviewer in *italic* and add our replies in **bold**. If we quote sentences from the manuscript, modified parts will be **bold**, while unmodified parts will be in normal font.

We have submitted a revised manuscript and **added a Supplement** with the data as requested.

**Referee #1**

**https://doi.org/10.5194/acp-2021-857-RC1**

*The manuscript by Mühle et al. investigates the relationship between calculated global emissions of PFC-318 and estimates of HCFC-22 production used as feedstock for production of fluorinated ethylene products used to manufacture Teflon. This manuscript extends previous work (Mühle et al., ACP, 2019) by updating several years of PFC-318 emission and by looking in more detail at the use of HCFC-22 as a feedstock. The material is well-presented, the underlying data on PFC-318 temporal variation is excellent, and the model used to evaluate emissions has been applied successfully to other AGAGE data.*

**We thank the reviewer for their overall positive evaluation of our manuscript.**

*I am not sufficiently expert to evaluate the accuracy of the HCFC-22 feedstock production data, which is the other significant compilation of this manuscript. Any errors in these reported production numbers will impact the conclusion, and, if relevant, the authors should comment on the reliability of these reports (especially since inconsistencies are noted for other gases regulated by the Montreal Protocol).*

**The large uncertainties we point out in the introduction are with respect to reporting of greenhouse gas emissions (e.g., PFC-318 ($c$-C$_4$F$_8$)) to UNFCCC, not with respect to reporting of production data of ozone depleting substances (e.g., HCFC-22) to the United Nations Environmental Programme (UNEP) under Article 7 of the MP. The latter production data are likely more accurate and complete, though estimates for the last year or two are often adjusted in the next report. First, production data of HCFC-22 at a limited number of production facilities can be relatively easily recorded and aggregated. In contrast, estimates of emissions from a whole country, e.g., of HCFC-22 resulting from manufacturing, use in and end of life of A/C systems, fugitive emissions at production facilities, etc. are much more difficult to estimate with accounting methods. Second, both developing and developed countries report their aggregated HCFC-22 production data to UNEP, while only developed countries need to report estimated greenhouse gas emissions to UNFCCC. We make the latter point in the Introduction and we explain in Section 2.2. that "**HCFC-22 production data for ... developing ... and ... developed ... countries were used here"**. We now also modified the previous sentence stating that "** ... various types of data, including FS production, are reported by **all** countries ... to ... UNEP ...**" to make it clear that all countries report to UNEP (in contrast to UNFCCC reporting). We have also added the caveat that "Data for the last year or two are often adjusted in the next report" in Section 2.2.**

*The major contribution of this paper is the observed linear relationship between HCFC-22 feedstock production and PFC-318 emission rates after about 2002. Attribution of the increasing PFC-318 emission due to industrial activity in China is also interesting.*

**We thank the reviewer for this positive assessment.**

*However, I thought the manuscript could have done a better job in describing why the relationship observed after about 2002 or so, is different from the preceding 10 years, and what was contributing to PFC-318 emissions prior to 1990. The unknown nature of the period prior to 2002 adds a level of uncertainty to the analysis that is not discussed.*

**Even though our manuscript is focused on more recent emissions, we have added the following information with respect to the large PFC-318 (c-$C_4F_8$) emissions prior to the 1990s in Section 3: "Historic $c$-$C_4F_8$ EFs were probably much higher, particularly during the early decades of PTFE production (1950-1990) when process controls or abatement were likely not in place. From the 1980s onwards, it is likely that EFs steadily improved with the advent of UNFCCC emission reporting requirements in the 1990s, concerns about the environment, climate change and product stewardship, abatement, and perhaps collection of $c$-$C_4F_8$ for use in the semiconductor industry, where it can be easily abated (Mühle et al., 2019, David Sherry, personal communication, 2022)." This was implicitly confirmed by a representative of one of the large U.S. producers of HCFC-22, PTFE and related derivatives. David Sherry (Nolan Sherry & Associates), who is an expert in the field of industrial scale chloro- and fluorochemical processes, agrees that this is very plausible, e.g., the then sole U.S. producer … with (P)TFE and derivatives was under intense pressure from the 1980s onwards to reduce its emission profile.**

*Between 1990 and 2002, the calculated PFC-318 emission rate remained nearly constant (even decreasing early in the period) while cumulative HCFC-22 FS production totaled about 1500 Gg.*

*For approximately the same HCFC-22 FS production from 2005-2010 (1405 Gg), the calculated PFC emission rate increased from 0.96 to 1.43 Gg/yr.*

**We agree that 1990-2002 cumulative global HCFC-22 FS production was ~1500 Gg (1525 Gg) and that PFC-318 ($c$-$C_4F_8$) emissions remained at a similar level (0.78 ± 0.06 Gg/yr), but we are not sure how the reviewer derived the 2005-2010 cumulative HCFC-22 FS production of 1405 Gg. Looking at Table 1, 2005-2010 cumulative global HCFC-22 FS production were 1939.2 Gg, ~30% larger than in the previous period.**

**As detailed above and now added to the manuscript in Section 3, our explanation for the declining PFC-318 ($c$-$C_4F_8$) emissions in the 1980s (see Mühle et al., 2019) and early 1990s was the result of overall emission reductions from the HCFC-22 TFE/HFP route in developing (A5) countries due to process improvements and regulatory pressure. Coincidently or not, the steady rise of $c$-$C_4F_8$ emissions with an EF of ~0.003 kg $c$-$C_4F_8$ emitted per kg of HCFC-22 produced for FS uses began in the early 2000s, when a) HCFC-22 FS production in developed countries ceased to increase and b) HCFC-22 FS production in developing countries started to rise and dominate the growth in total global HCFC-22 FS production. Due to the relatively stable HCFC-22 FS production in developed countries, we cannot determine from our correlation analysis whether current PFC-318 ($c$-$C_4F_8$) EFs in developing (A5) or developed (non-A5) countries are similar or not, which we now clearly state in the revised manuscript in Section 3 and in the Summary and Conclusions Section. Atmospheric measurements covering individual countries and facilities are needed to determine this.**

*Similarly, the last part of the record shows constant HCFC-22 FS production but continued increase in PFC-318 emission. How is this rationalized?*

**We must assume that the reviewer is referring to HCFC-22 FS production in developed countries (non-A5), as that is the only record which is roughly constant in recent years. However, what really matters is that global (A5 + non-A5) HCFC-22 FS production has been rising all throughout the record as shown in Figure 2, driven by the increase of production in developing countries (mostly China).**

*I would like to understand the different emission response of PFC-318 to HCFC-22 production to have better confidence in the quantitative results presented. The correlation presented looks compelling but deserves more detailed analysis of potential factors that might contribute to the correlation.*

**We hope that the explanations above and the additions and revisions, particularly the added discussion of the historic $c$-$C_4F_8$ emissions (Section 3) and the improved discussion of the 1996–2001 period with little HCFC-22 production in developing countries (Section 3), allow readers to better understand factors that affect the relationship between $c$-$C_4F_8$ emissions of HCFC-22 FS production.**

*It might have also been interesting, if the data is available, to discuss the relative importance of HCFC-22 FS production to other sources of HCFC-22 and to the global burden and trend of HCFC-22.*

**We are not sure if we fully understand the question. While total production and emissions of HCFC-22 are important topics, e.g., to understand emissions of HFC-23 by-product and to understand the impact of HCFC-22 emissions on climate and the ozone layer, a discussion of "the relative importance of HCFC-22 FS production to other sources of HCFC-22 and the global burned and trend of HCFC-22" are not directly relevant in the context of this paper. HCFC-22 that is produced for feedstock (FS) applications, i.e., mostly to produce PTFE and related fluoropolymers and fluorochemicals (via TFE and HFP), will be (almost completely) destroyed. Therefore, it will not be emitted into the atmosphere (to a significant degree) and has no (little) influence on the global burden and trend of HCFC-22. Only HCFC-22 that is produced for dispersive uses (non-FS uses), such as refrigeration or foam blowing, will be immediately or eventually emitted into the atmosphere (unless destroyed/recovered), determining the global burden and trend of HCFC-22 in the atmosphere. However, HCFC-22 that is emitted into the atmosphere from these dispersive, that is non-FS, uses is not relevant for the emissions of PFC-318 ($c$-$C_4F_8$) as it is not pyrolyzed to TFE/HFP and therefore does not lead to $c$-$C_4F_8$ by-product emissions.**

**To clarify this, we added the following paragraph to Section 2.2: "Note, that we do not discuss HCFC-22 non-FS production statistics, i.e., HCFC-22 produced for emissive uses (e.g., refrigeration and foam blowing). While critical for understanding HCFC-22 emissions and HCFC-22 atmospheric burden, amounts of HCFC-22 produced for non-FS uses are not relevant for $c$-$C_4F_8$ emissions. We also do not discuss total HCFC-22 (non-FS plus FS) production. While critical for understanding HFC-23 by-product emissions (from total HCFC-22 production) and HFC-23 atmospheric burden, they are not directly relevant for $c$-$C_4F_8$ emission studies. Only HCFC-22 that is produced for FS uses and pyrolyzed to TFE/HFP with $c$-$C_4F_8$ by-product is relevant for $c$-$C_4F_8$ emissions and $c$-$C_4F_8$ atmospheric burden."**

**We also clarified that HFC-23 is "**known to be a by-product of **total (FS and non-FS)** production of HCFC-22 …" **in the 2[nd] to last paragraph of the introduction and shortened this sentence to avoid repetition.**

*AGAGE data shows a consistent linear increase in PFC-318, while the rate of HCFC-22 increase has slowed. Does this mean that HCFC-22 FS emissions are becoming a more significant component of ambient HCFC-22 relative to other changing sources?*

We are not sure if we fully understand the question but believe that there is a misunderstanding. The rate of HCFC-22 increase has slowed down due to the MP-mandated phase-out of production and consumption of HCFC-22 for emissive uses. In this paper we present evidence that the PFC-318 ($c$-$C_4F_8$) increase in the atmosphere is related to HCFC-22 feedstock (FS) production - as most of that HCFC-22 FS is pyrolyzed to TFE and HFP in a process that causes PFC-318 by-product emissions. While there may be minor fugitive emissions of HCFC-22 that is being produced for HCFC-22 FS applications, almost all of it is destroyed in FS applications and converted into other chemicals, such as TFE and HFP. Global HCFC-22 emissions and atmospheric burden are determined by the production of HCFC-22 for emission (non-FS) uses, such as refrigerant and foam blowing applications, and their immediate or eventual release therefrom to the atmosphere. We hope that our improvements to the revised manuscript in Section 2.2 now make this clear.

However, we would like to add that due to the phase-out of ozone depleting substances, such as HCFC-22, for dispersive uses (MP controls are 100% reduction of production and consumption by 2020 in developed countries, 35% reduction by 2020, 67.5% reduction by 2025, and 100% reduction by 2030 in developing countries) and the growing demand for HCFC-22 for FS uses (e.g., to produce PTFE), which is not regulated by the MP, the fraction of HCFC-22 production for dispersive uses (non-FS) versus FS uses has declined substantially, while the total amount of HCFC-22 produced (FS + non-FS) has continued to increase.

*In a similar vein, the authors mention $CHF_3$ associated with the production of HCFC-22, and thus it has a source related to PFC-318 emission. Can temporal trends in $CHF_3$ vs PFC-318 observed at AGAGE sites provide additional insight into the relative importance of the different source emissions over time? Data from at least one of the AGAGE sites (THD) seems to show a change in slope of the relationship between $CHF_3$ and PFC-318 occurring around 2015. It seems a shame not to get better use of the excellent measurements from the AGAGE sites. This may well be beyond the scope of the manuscript intended by the authors, but it could make the manuscript more interesting.*

The relationship between total HCFC-22 produced (FS + non-FS) and HFC-23 by-product emitted is very complicated due to different degrees of process optimization (and thus HFC-23 formation), HFC-23 recycling and abatement and thus HFC-23 by-product emissions with time-varying and factory specific emission factors (and minor use of HFC-23 as refrigerant) and atmospheric abundance. It is much more straightforward to use the available UNEP HCFC-22 FS production data as only this portion of total HCFC-22 production is relevant for PFC-318 ($c$-$C_4F_8$) by-product emissions. We hope that our improvements to the revised manuscript, particularly the corresponding explanation added to Section 2.2, make this now clear. With respect to the intricacies of HCFC-22 production, emissions, and atmospheric burden as well as HFC-23 by-product emissions and atmospheric burden, we kindly refer the reviewer to AGAGE and NOAA publications on this matter (e.g., Stanley et al., Nature Commun., doi:10.1038/s41467-019-13899-4, 2020; Simmonds et al., Atmos. Chem. Phys., doi:10.5194/acp-18-4153-2018, 2018; Montzka et al., Geophys. Res. Lett., doi:10.1029/2009gl041195, 2010; Miller et al., Atmos. Chem. Phys., doi:10.5194/acp-10-7875-2010, 2010) as well as the discussions in recent (2018) and the upcoming (2022) WMO Scientific Assessment of Ozone Depletion.

*Other minor comments/questions:*

*Line 84: Given the locations of the sites used for the 12 box model, it brings up the question of how much impact was observed in the pollution events that are removed from the data set for further analysis? And, if significant, could further analysis of these pollution events provide further insight into regional or*

*long-range sources? Though not critical, it would be interesting to know the fraction of data that were removed due to pollution filtering.*

**Indeed, the analysis of pollution events above background mixing ratios at polluted sites provides valuable information about regional emissions. However, for the baseline stations used here to derive global emissions, only infrequent and small PFC-318 ($c$-C$_4$F$_8$) pollution events were observed, and the pollution filter retained most observations: 99.1% at Mace Head (MHD), 99.6% at Trinidad Head (THD), 100.0% at Ragged Point (RPG), 100.0% at American Samoa (SMO), and 99.7% at Cape Grim (CGO). Correspondingly, using data from MHD, Tacolneston, UK, Jungfraujoch, Switzerland, and Monte Cimone, Italy, Mühle et al. (2019) estimated only very minor $c$-C$_4$F$_8$ emissions of 0.026 ± 0.013 Gg/yr from northwestern Europe (2013– 2017) without any significant temporal trend, ~1% of global emissions. Even smaller emissions were estimated from Australia based on CGO data.**

**In contrast, as detailed in Mühle et al. (2019), the Gosan station on Jeju Island, South Korea, observes "by far the most frequent and most pronounced pollution events … above the NH background" in the AGAGE network. Only 47% of the observation at Gosan were classified as non-polluted. The data was used to infer significant regional emissions of $c$-C$_4$F$_8$ in East Asia, dominated by emissions from (Eastern) China. Emission from Russia and India (aircraft campaign) were likely smaller but also significant. The spatial patterns of $c$-C$_4$F$_8$ emissions were consistent with emissions from the production of PTFE and related fluoropolymers via HCFC-22 pyrolysis, which led to the current study.**

**As the current manuscript is focused on global $c$-C$_4$F$_8$ emissions from baseline station data and the relationship with HCFC-22 FS production, we do not think that we should repeat these results. However, as two reviewers asked similar questions, we added this brief statement in Section 2.1: "For these baseline stations more than 99% of the data were retained, reflecting minor regional $c$-C$_4$F$_8$ emissions near these stations. In contrast, in East Asia strong and frequent pollution events were observed and corresponding strong emissions were inferred, as detailed in Mühle et al. (2019)."**

*Line 110: The authors note that the results here agree with Mühle et al., 2019. Aren't the data, model, and methods identical between this study and earlier? I thought only the last few years of data were new.*

**The reviewer is correct, the emissions presented here were derived with a mostly identical, albeit longer, AGAGE data set, and inverse method as used by Mühle et al. (2019). Still, we think that it is important to point out that there is close agreement of the global $c$-C$_4$F$_8$ emissions presented in the earlier work with those presented here (until 2020). However, we agree that the wording was perhaps not ideal, so we revised the first and second sentence in Section 3 to make this clear: "O**ur updated global inversion results show that $c$-C$_4$F$_8$ emissions were … . However, … with a relatively steady increase to 2.26 Gg yr-1 in 2017 (Table 1 and Fig. 2, black diamonds**; these emissions are very similar those in Mühle et al. (2019), which were based on a mostly identical, albeit shorter duration, AGAGE data set and inverse method**)."

*Figure 2: The black diamond curve is not identified in the caption or figure as the PFC-318 emission rate.*

**Thank you for pointing this out as we did not notice that this legend entry disappeared during copying and pasting. We fixed this in the revised manuscript and the black diamonds are now defined as "Global $c$-C$_4$F$_8$ emissions".**

**Referee #2**

**https://doi.org/10.5194/acp-2021-857-RC2**

*This manuscript reports a decade of measurements of perfluorocyclobutane (c-C₄F₈) from the remote AGAGE sites. c-C₄F₈ is a long-lived potentially important greenhouse gas. The measurements accurately define emissions, indicating they are clearly northern hemisphere and correlated with the production of HCFC-22. The analysis shows that c-C₄F₈ is a likely by-product and from the observations they calculate an emission factor. The authors argue that better process management could reduce these fugitive emissions, which are much larger than the direct intentional production reported to the UNFCCC. This work is an important extension to the Mühle et al. (2019) work on this gas.*

*The manuscript is well written (except see below) and provides a valuable contribution to our understanding of the synthetic greenhouse gases and their potential role in climate change.*

**We thank the reviewer for their overall positive evaluation of our manuscript.**

*The authors need to focus more on what is new here and not confuse with A5, non-A5, and China (3 different entities?).*

**Following the reviewer's advice (also detailed below), we now begin the Summary and Conclusions Section with our new results, and we reduced the discussion of previously published work.**

**We also realized that we need to make it clearer that China is one of the developing (A5) countries. As detailed below we have modified the Table 1 header for China, the caption for Table 1 and the text in Section 2.2 (end of the first paragraph) and Section 3 (last paragraph) to clearly explain that most of the HCFC-22 feedstock (FS) production in developing (A5) countries occurs in China.**

**We also now focus the correlation analysis on global HCFC-22 FS production versus global $c$-C₄F₈ emissions. We have significantly shortened the discussion of the correlation between HCFC-22 FS production in developing (A5) countries versus global $c$-C₄F₈ emissions with the conclusion that we cannot determine if current emission factors in developed and developing countries are similar or not.**

**Instead of using the technical terms "non-A5 countries" and "A5 countries" alone, we have revised the manuscript to use "developed (non-A5) countries" and "developing (A5) countries" or "developed countries" and "developing countries" as we hope that this improves overall clarity and readability.**

**Lastly, as detailed further below, we have revised the analysis of the emission factor for China at the end of Section 3.**

**We believe that the revised manuscript flows more naturally and is easier to understand.**

*Further, the data archive must be upgraded before publication.*

**We would kindly like to point out that the archive data is already available as Supplement to Mühle et al. (2019) https://acp.copernicus.org/articles/19/10335/2019/acp-19-10335-2019-supplement.zip. We have, however, deposited the archive data and the monthly mean baseline in-situ measurement data in a Supplement to this manuscript. We also realized that the use of archive data should have been**

**better explained. While we do not want to repeat all the information from Mühle et al. (2019), we modified the description of Fig. 1 in the second paragraph of Section 2.1 and added a sentence explaining the use of archive data:** "Fig. 1 shows the continued increase of pollution free monthly mean $c$-$C_4F_8$ mole fractions in the global atmosphere **since the start of in-situ measurements. Atmospheric abundances before in-situ measurements were reconstructed based on measurements of samples of the Cape Grim Air Archive (CGAA) for the extratropical Southern Hemisphere and archived air samples from various sources for the extratropical Northern Hemisphere (not shown) as detailed in Mühle et al. (2019)."** We hope that this is now sufficiently explained.**

*L30 – Glad to see a clear chemical definition, noting the range of names used to describe $c$-$C_4F_8$.*

**Indeed.**

*L84 – Can you comment on whether the pollution events that were removed contained high levels of $c$-$C_4F_8$? Thus indicating nearby production? Which stations? This might be useful.*

**The reviewer brings up an important point. As detailed in our reply to Referee #1, the analysis of pollution events above background mixing ratios at polluted sites provides valuable information about regional emissions. However, for the baseline stations used here to derive global emissions, only infrequent and very small $c$-$C_4F_8$ pollution events were observed, and the pollution filter retained most of the observations (>=99%). Correspondingly, Mühle et al. (2019) estimated only very minor $c$-$C_4F_8$ emissions of 0.026 ± 0.013 Gg/yr from Central Europe, ~1% of global emissions. Even smaller emissions were estimated for Australia based on Cape Grim (CGO) data. In contrast, the frequent and strong pollution events at Gosan, South Korea allowed Mühle et al. (2019) to infer significant regional emissions of $c$-$C_4F_8$ in East Asia, dominated by emissions from (Eastern) China. Emission from Russia and India (aircraft campaign) were likely smaller but also significant. The spatial patterns of $c$-$C_4F_8$ emissions were consistent with emissions from the production of PTFE and related fluoropolymers via HCFC-22 pyrolysis, which led to the current study.**

**As the current manuscript is focused on global $c$-$C_4F_8$ emissions from baseline station data and the relationship with HCFC-22 FS production, we do not think that we should repeat these results. However, as two reviewers asked a similar question, we added this brief statement:** "For these baseline stations more than 99% of the data were retained, reflecting minor regional $c$-$C_4F_8$ emissions near these stations. In contrast, in East Asia strong and frequent pollution events were observed and corresponding strong emissions were inferred, as detailed in Mühle et al. (2019)."

*L120 – It would be good to include the black diamonds in Fig 2 in the legend, making it clear that they are the global $c$-$C_4F_8$ emissions.*

**We thank the reviewer for noticing that the legend for global $c$-$C_4F_8$ emissions had disappeared upon inserting the figure into Word. We have fixed this.**

*BTW, is it clear that China is neither A5 nor non-A5? Is it being double counted here and in the Table? This is confusing.*

**We apologize for not making it clearer that China is considered a developing (A5) country, hence, the "China only" HCFC-22 feedstock (FS) production are included in the developing (A5) countries totals.**

**We now clarify this in the text by adding in Section 2.2 at the end of the fist paragraph: "Table 1 shows that Chinese HCFC-22 FS production from 2008 to 2018 accounted for (84 ± 6)% of A5 (developing countries) HCFC-22 FS production ((86 ± 3)% if the last year, 2018, is excluded), i.e., most of the HCFC-22 feedstock (FS) production in developing (A5) countries occurs in China.". We also added a similar statement to the Table 1 caption and modified the header for the China column to "A5** China only**". We hope that this is clear now.**

*L135 – I do not understand the effort at linear fitting in Fig. 3. It is confusing. The TOTAL (green) fit makes some sense, presumably implying a source of ~0.14 Gg/y that is from non-HCFC-22 sources, and your yield of 0.31% (kg/kg). The blue fit of global to only A5 emissions does not make sense. The explanation in L126-155 does not help. Why do you come up with an emission factor assuming that only A5 countries have processes that emit $c$-$C_4F_8$. If this is the case, then why fit to global? Please introduce the logic of this approach if you think it relevant.*

**We largely agree with the reviewer. Our intention had been to demonstrate that we derive very similar emissions factors (EFs) whether we take global HCFC-22 FS production into account or only consider HCFC-22 FS in developing countries. The primary reason is that HCFC-22 FS production in developed countries was essentially flat since 2002 and therefore has only a minor influence on the derived EF (slope). We had hoped that we could determine whether current emission factors in developing and developing countries are similar or not, but we cannot do this.**

**Following this reviewer's advice, we now begin with and focus on the correlation between global HCFC-22 FS production and global $c$-$C_4F_8$ emissions. We rephrased and shortened the discussion of the also observed good correlation between HCFC-22 FS production in developing countries and $c$-$C_4F_8$ emissions, mostly due to the lack of change in HCFC-22 FS production in developed countries, and end by stating that we cannot determine whether current $c$-$C_4F_8$ EFs in developing and developed countries are similar or not.**

*L174ff – Again, this is confusing with the separation of China and comparing it the A5 and non-A5. Discussing China, and then eastern China is more confusing as to what is measured and what is known.*

**As indicated above, we have revised the section on $c$-$C_4F_8$ emissions from China. We now begin by explaining that this analysis is based on the HCFC-22 FS production data for China (Table 1) and the emissions reported by Mühle et al. (2019). We then explain that this discussion is of interest because a major fraction of HCFC-22 FS production in developing countries occurs in China. We then present the caveat that the underlying atmospheric measurements were mostly sensitive to emissions in Eastern China, which means that emissions from several production complexes in other parts of China with likely $c$-$C_4F_8$ emissions are probably missing. We refer to Fig. 7 in Mühle et al. (2019) and added a table with production complexes in China to the Supplement indicating which are within the "Eastern China" emission estimate footprint of Mühle et al. (2019). Not surprisingly, we find an emission factor for China which appears too low, likely due to the missing emissions from facilities in other parts of China not covered by atmospheric measurements. We conclude that more measurements are needed to investigate this. We hope that the revised text flows more naturally and is easier to understand.**

*L181-201 – This conclusion section is again very confusing for this reviewer. It mixes discussion of the processes in great detail with the regions. It brings in A5 and non-A5 and then discussions Eastern China, China, and Russia, without any understanding as to why these are important. I think the authors want to*

*emphasize the Muhle et al (2019) regional attribution with this global one. I would de-emphasize the previously published work and focus on the A5 and non-A5. Then separately discuss China if you want but do not confuse with the A5 and non-A5 modeling here.*

**Following the reviewer's advice, we shortened and streamlined the Summary and Conclusions and put less emphasis on previously published work. We now begin with our new results of continued *c*-C₄F₈ emission increases, followed by our conclusion that the "industrial scale HCFC-22 pyrolysis" leads to the observed correlation between global *c*-C₄F₈ emissions and global HCFC-22 FS production. We added that "Based on the available atmospheric measurements we cannot determine whether current EFs for developed … and developing … countries are similar or dissimilar". We have removed the sentence on emissions from China here (but improved the corresponding discussion in Section 3). The general conclusion that more atmospheric measurements are needed remains. Several other improvements resulted from comments of all reviewers.**

*L202-206 – This is a clear summary, the above is confusing.*

**Thank you. We have left this last paragraph largely as it was and believe that the revised, shortened and streamlined beginning of the Summary and Conclusions section is now clear as well.**

*\*L210 – I am afraid that the data availability is really inadequate and so 'old'. You must post on a DOI the datasets used here, particularly Table 1 and Figures 2 and 3 so that others can compare readily with this work. While AGAGE seems to live forever, it is not appropriate to list your data from an non-ODI source like your own website. Please use a regular doi, I am that MIT must have one.*

**We understand the reviewer's concern and have deposited the data used as a Supplement to this manuscript so that it will have a unique doi. Most up-to-date and quality-controlled AGAGE data are available at [https://agage.mit.edu/data](https://agage.mit.edu/data) ([http://agage.eas.gatech.edu/data_archive/agage/gc-ms-medusa/complete/](http://agage.eas.gatech.edu/data_archive/agage/gc-ms-medusa/complete/), [http://agage.eas.gatech.edu/data_archive/agage/gc-ms-medusa/monthly/](http://agage.eas.gatech.edu/data_archive/agage/gc-ms-medusa/monthly/)) and/or upon request. AGAGE data are also regularly submitted to [https://data.ess-dive.lbl.gov/data](https://data.ess-dive.lbl.gov/data); at the time of writing, the most recent AGAGE data are available at [https://data.ess-dive.lbl.gov/view/doi:10.15485/1841748](https://data.ess-dive.lbl.gov/view/doi:10.15485/1841748). We have modified the sentence accordingly and added the used data and this information to the Supplement.**

**Referee #3**

**https://doi.org/10.5194/acp-2021-857-RC3**

*General Comments:*

*This paper represents an important advance in our understanding of the sources of global perfluorocyclobutane emissions. These emissions matter. As discussed in this paper and in Mühle et al (2019), perfluorocyclobutane is a long-lived, potent synthetic GHG whose emissions are rapidly growing. Other data sources also indicate that perfluorocyclobutane emissions are significant. Data from the US Inventory of GHG Emissions and Sinks and from the USEPA Greenhouse Gas Reporting Program indicate that perfluorocyclobutane is the third most emitted perfluorocarbon in the US in GWP-weighted terms, behind perfluoromethane and perfluoroethane. Perfluorocyclobutane is also consistently among the top two or three most emitted fluorinated GHGs of any kind from US fluorochemical production in GWP-weighted terms.*

*The data and analysis presented in the paper make a compelling case that pyrolysis of HCFC-22 is the chief source of perfluorobutane emissions globally. This case begins with the detailed and richly sourced discussion of the pyrolysis process and the role of perfluorocyclobutane in it and proceeds through the author's calculations of global emissions of perfluorocyclobutane and discussion of HCFC-22 feedstock production in developed and developing countries. The authors' representation of the pyrolysis process and its generation of perfluorocyclobutane is consistent with discussions that this reviewer has had with US producers of HFP and TFE. Their analysis of UNEP and TEAP reports of HCFC-22 production for feedstock use, and the correlations between this production and perfluorocyclobutane emissions, is thorough and thoughtful. Their interpretation of these trends and relationships is persuasive but nuanced.*

**We thank the reviewer for their very positive overall assessment.**

*Specific Comments:*

*To put the inferred global $c\text{-}C_4F_8$ emissions into perspective, it would be helpful to compare them briefly to inferred global emissions of other long-lived GHGs, such as $SF_6$, $CF_4$, and $C_2F_6$, both in unweighted and GWP-weighted terms.*

**Based on AGAGE data, 2020 global emissions of perfluorinated compounds (PFCs), $SF_6$ and $NF_3$ were ~13.7 ($CF_4$), ~2.15 ($C_2F_6$), ~0.58 ($C_3F_6$), ~2.32 ($c\text{-}C_4F_8$), ~8.91 ($SF_6$) and ~2.86 ($NF_3$) Gg/yr. Using AR6 $GWP_{100}$ this equates to 101 ($CF_4$), 27 ($C_2F_6$), 5 ($C_3F_8$), 24 ($c\text{-}C_4F_8$), 225 ($SF_6$), and 50 ($NF_3$) million metric tons of $CO_2$-eq/yr. We have added the 2020 $c\text{-}C_4F_8$ $CO_2$-eq. emissions to the manuscript but have added the information for other PFCs, $SF_6$ and $NF_3$ to the Supplement as they were not directly determined for this manuscript.**

*On page 2, line 35, recommend deleting "regulated and" and "the Kyoto Protocol of," resulting in the following sentence: "Emissions of $c\text{-}C_4F_8$ from developed countries are reported under the United Nations Framework Convention on Climate Change (UNFCCC)." The requirement to report emissions of $c\text{-}C_4F_8$ is*

*what is relevant here, and it applies to countries that are signatories to the UNFCCC but not the Kyoto Protocol.*

**We thank the reviewer for this correction and have implemented it as suggested.**

*On page 2, line 58, the authors mention electrochemical fluorination (ECF) as a potential alternative means for manufacturing TFE and HFP that would generate less waste than the currently dominant method. This was surprising, because at least some types of ECF are known to generate large quantities of waste products. This reviewer is not familiar with the research cited by the authors in support of their suggestion to explore ECF as an alternative (Ebnesajjad and Mierdel), which may be based on more recent and refined types of ECF. I recommend qualifying the recommendation, perhaps by inserting "refined methods of" before "electrochemical fluorination" on line 58 and in other places where ECF is mentioned.*

**We thank the reviewer for their insights into ECF. Indeed, Mierdel et al. (2019) discuss the problem of the "low selectivity to perfluorinated products obtained" by ECF and that "up to 70% of the product spectrum consists of only partly fluorinated alkanes". Using the example of HFC-23 as a partly fluorinated alkane (which could also be obtained from a conventional HCFC-22 production waste stream), the authors then proceed to discuss how such partly fluorinated alkanes can be pyrolyzed into TFE and HFP. They conclude that ECF followed with their suggested pyrolysis of partly fluorinated alkanes (such as HFC-23) could result in "tremendous saving potential in energy consumption, undesirable by-products, disposing waste materials, and the complete substitution of chlorine chemistry" and that "the developed process is suitable to be transferred into the industrial scale". Ebnesajjad (2015) summarizes research from three patents and cites as one of the benefits the "potential to operate in a closed-loop". Based on these discussions and following the reviewer's suggestion, we modified our statements on ECF to "refined methods of electrochemical fluorination and waste recycling" in the Abstract, Introduction, and the Summary and Conclusion.**

*Figure 2 on page 6 contains a wealth of interesting data that is generally thoroughly discussed in the paper. Particularly fascinating was the analysis of using the 1996 through 2000 feedstock and emissions data to estimate an emission factor for developed countries, where the authors concluded that the current emission factor from developed countries was probably lower than the average from that period.*

**We thank the reviewer for this positive assessment but would like to note that we have shortened the discussion of the 1996-2002 EF for develop countries as we thought that it was unnecessarily long.**

*However, there was a sharp divergence between $c\text{-}C_4F_8$ emissions and global feedstock production in 2009 that the authors did not mention. This is sufficiently striking that it is probably worth discussing briefly, even if the conclusion is that the cause of the divergence is not known. (The same is true of a smaller divergence in 2016.)*

**This period is very difficult to analyze as the global HCFC-22 market was very disrupted. The decrease in HCFC-22 FS production in 2009 (developed countries and total global) was preceded by a large increase in HCFC-22 FS production in developing countries in 2008 (Table 1 and Fig. 2). This was a**

result of increased Chinese HCFC-22 production for demand-based FS uses, most notably PTFE, which may have displaced exports into China. Outside of China, there was also a shortage of hydrogen fluoride, needed to produce HCFC-22 and almost all other fluorocarbons (David Sherry, personal communication, 2022). It is also possible that some of the HCFC-22 FS produced at the year-end was used (pyrolyzed) in the next year, explaining the smooth $c$-$C_4F_8$ emission history. As both Reviewers 3 and 4 have asked a similar question, we have added corresponding paragraph to the revised manuscript in Section 2.2, but we have refrained from adding this in the "Results and Discussion" section.

As Table 1 indicates, the 2015/2016 dip in HCFC-22 FS production (China and A5 countries) was partially compensated by an increase of HCFC-22 FS production in developed (non-A5) countries (e.g., in Japan, David Sherry, personal communication, 2022).

While we may not be able to explain all variability, the overall correlation between $c$-$C_4F_8$ and global HCFC-22 FS production is strong ($R^2$ = 0.97).

*On page 8, the authors state "Current industry knowledge is that less than 2% of HCFC-22 FS produced is used in reactions that do not involve the TFE/HFP/c-$C_4F_8$ route," but they do not cite a source. A source should be cited for this statement because the correlation between HCFC-22 feedstock production and TFE/HFP/c-$C_4F_8$ production is fundamental to the method used by the authors to reach their conclusions.*

We have revised this sentence and the preceding paragraph slightly based on consultations with David Sherry (Nolan Sherry & Associates) an expert in the field of industrial scale chloro- and fluorochemical processes and Andy Lindley (UNEP Medical and Chemicals Technical Options Committee, MCTOC). First, we state that "… **it is estimated that almost all (David Sherry, Andy Lindley, personal communications, 2022)** of global HCFC-22 FS production is used to product TFE and HFP, to in turn produce PTFE and related fluoropolymers and fluorochemicals, … ". Then we detail in the next paragraph that "Current estimates are that perhaps 3% of HCFC-22 FS produced is used in reactions other than the TFE/HFP route (David Sherry, Andy Lindley, personal communication, 2022) that is without $c$-$C_4F_8$ by-product; products include sulfentrazone herbicide, pantoprazole (acid reflux) pharmaceutical, isoflurane and desflurane anesthetics, as well as high-purity HFC-23 for refrigeration use and as feedstock to manufacture iodotrifluoromethane, halon-1301 and from this, fipronil pesticide, mefloquine (antimalarial) and DPP-IV inhibitor (antidiabetic) pharmaceuticals (TEAP, 2021)." We have also revised the statement in the "Summary and Conclusion" to "… almost all …".

*Technical Corrections*

*Page 2, line 55: "TFE producer" is missing an "s."*

**DONE.**

*Figure 2: The legend is missing an entry for the black diamonds, which appear to represent global inferred c-$C_4F_8$ emissions. In addition, the black diamonds appear to be offset from the other icons in the*

*chart, perhaps because the values shown are from the middle of the year. Whatever the reason for the offset, it should be explained.*

**We thank the reviewer for noticing these two issues, which we have fixed in the revised figure and manuscript.**

*Figure 3: The legend currently appears very close to the data, making it look as if the icons being explained in the legend are part of the data. Recommend either moving the legend or enclosing it in an outline so that the reader can distinguish between the legend and the data.*

**We have shrunk the legend. We also added descriptions of the symbols to the legend text.**

**Referee #4**

**https://doi.org/10.5194/acp-2021-857-RC4**

*This paper provides a valuable addition to the literature linking the pyrolysis of HCFC-22 feedstocks to c-$C_4F_8$ by-product emissions measured in the atmosphere. The paper is well-written. The explicit comparison to global feedstock supplies and derivation of a global emission factor is new and support the conclusion that pyrolysis of HCFC-22 is the dominant source of c-$C_4F_8$ in the atmosphere. The link to feedstock production in developing countries is interesting and provides a useful estimate of overall emission factor from the sector in developing countries.*

**We thank the reviewer for their very positive overall assessment.**

*Specific comments:*

*The paper would however benefit from discussing other sources of c-$C_4F_8$ in a quantitative manner if possible. The intercept from Figure 3 from global production of HCFC-22 feedstock suggests a significant other source. Emissions of c-$C_4F_8$ from semiconductor manufacturing, largely from consumption of c-$C_4F_8$ for etching processes, are small but not insignificant and could account for a large portion of the background. In 2018, WSC reported emissions of ~0.13 Gg (https://www.eusemiconductors.eu/sites/default/files/23rdWSCJoint-Statement_May2019Xiamen-TOC_FINAL.pdf). Total emissions of c-$C_4F_8$ from the electronics sector may be larger, as this estimate does not include emissions from PV, LDC or MEMS. Use of c-$C_4F_8$ in the semiconductor industry started increasing at about the same time as the increase of $C_4F_8$ in the atmosphere (early 2000s – see Francesca Illuzzi & Harry Thewissen (2010) Perfluorocompounds emission reduction by the semiconductor industry, Journal of Integrative Environmental Sciences, 7:sup1, 201-210, DOI: 10.1080/19438151003621417 ).*

**We thank the reviewer to provide these insights and urge us to revisit this topic. We have added to our discussion on the observed post-2001 correlation that "the correlation indicates an emission factor (EF) of (0.0031 +- 0.0001) kg $c$-$C_4F_8$ emitted per kg of HCFC-22 produced for FS use (to produce TFE/HEP), with an intercept of 0.14 Gg/yr $c$-$C_4F_8$, presumably reflecting $c$-$C_4F_8$ emissions from other sources, such as semiconductor (SC), photo voltaic (PV), liquid crystal display (LCD), and micro-electromechanical system (MEMS) production. The annual reports of the World Semiconductor Council (WSC) (http://www.semiconductorcouncil.org/public-documents/joint-statements-from-prior-wsc-meetings/) contain estimates of $c$-$C_4F_8$ emission from SC production in China, Taiwan, Europe, Japan, South Korea, and the United States. They range from ~0.05 Gg/yr in 2012-2014 to ~0.11 Gg/yr in 2018-2019, somewhat smaller than the 0.14 Gg/yr intercept. We also updated the global $c$-$C_4F_8$ bottom-up inventory estimate from Mühle et al. (2019) using the 2021 National Inventory Submissions to UNFCCC (https://unfccc.int/ghg-inventories-annex-i-parties/2021) and then augmented this with their top-down emission estimates for Western Japan, South Korea, North Korea and Taiwan (but not China). The resulting emission estimates are ~0.09 Gg/yr in 2012-2019 and include top-down $c$-$C_4F_8$ emission estimates from all processes such as SC, PV, LCD, and MEMS production in these four countries, but also from any HCFC-22 FS pyrolysis in these countries, most notably in Japan. We did not include U.S. EPA emission estimates of ~0.06 Gg/yr $c$-$C_4F_8$ from U.S. fluorinated gas producers**

(https://www.epa.gov/ghgreporting/data-sets) in this updated estimate, as most of these $c$-$C_4F_8$ emissions stem from facilities that pyrolyze HCFC-22 (Deborah Ottinger, personal communication, 2022). Overall, the data support our conclusion that currently $c$-$C_4F_8$ emissions from sources other than HCFC-22 FS use (to produce TFE/HFP) are small, perhaps ~0.1-0.14 Gg/yr".

We have also added the following discussion on historic $c$-$C_4F_8$ emissions to the manuscript just after our discussion of the "… global EF … are similar to the optimal production conditions …" in Section 3: "Historic $c$-$C_4F_8$ EFs were probably much higher, particularly during the early decades of PTFE production (1950-1990) when process controls or abatement were likely not in place. From the 1980s onwards, it is likely that EFs steadily improved with the advent of UNFCCC emission reporting requirements in the 1990s, concerns about the environment, climate change and product stewardship, abatement, and perhaps collection of $c$-$C_4F_8$ for use in the semiconductor industry, where it can be easily abated (Mühle et al., 2019, David Sherry, personal communication, 2022)."

We thank the reviewer for encouraging us to "*discuss… other sources of c-C₄F₈ in a quantitative manner if possible*" and think that these added discussions have improved the revised manuscript.

*On page 8 of this manuscript, you refer to the possibility of significant emissions from the semiconductor industry between 1996 and 2001, but this seems unlikely.  No emissions of C₄F₈ were reported to the US EPA prior to 2002.*

Considering the revised and expanded discussion of other $c$-$C_4F_8$ sources, we essentially agree with the reviewer. We have removed this statement and shortened this section as we thought that it was unnecessary long.

*Considering that there was a large dip in production of HCFC-22 in 2009 but an increase in emissions, this may be evidence that the by-product emissions from Annex 5 countries and China is greater than for developed countries, as these two regions had only a minor decrease or an increase (China) in 2009 or alternatively that there was not a corresponding drop in production for other c-C₄F₈ sources.*

This period is difficult to analyze as the global HCFC-22 market was very disrupted. As we have detailed in our response to Reviewer 3, the decrease in HCFC-22 FS production in 2009 (developed countries and total global) was preceded by a large increase in Chinese HCFC-22 FS production in developing countries in 2008 (Table 1 and Fig. 2). This was a result of increased Chinese HCFC-22 production for demand-based FS uses, most notably PTFE, which may have displaced exports into China. Outside of China, there was also a shortage of hydrogen fluoride, needed to produce HCFC-22 and almost all other fluorocarbons (David Sherry, personal communication, 2022). It is also possible that some of the HCFC-22 FS produced at the year-end was used (pyrolyzed) in the next year, explaining the smooth $c$-$C_4F_8$ emission history. As both Reviewers 3 and 4 have asked a similar question, we have added a corresponding paragraph to the revised manuscript in Section 2.2, but we have refrained from adding this in the "Results and Discussion" section.

*There was also a dip in production in 2015 (China + A5) but no corresponding dip in emissions.  Appears to instead be large increase.  There was a corresponding increase in c-C₄F₈ emissions in S. Korea in your*

*previous paper (Muhle 2019) and in US emissions from the semiconductor industry (EPA 2020). Even if the authors are not sure of the source of these differences in trends, it would be useful to discuss the possible sources.*

**As Table 1 indicates, the 2015 dip in HCFC-22 FS production (China and A5 countries) was partially made up for by an increase of HCFC-22 FS production in developed (non-A5) countries (e.g., in Japan, David Sherry, personal communication, 2022). The enhancement of emissions from South Korea in 2014 and 2015 (~0.04 Gg/yr) compared to other years (0.01–0.02 Gg/yr) was relatively small in comparison to global $c$-C$_4$F$_8$ emissions. Similarly, the most recent U.S. EPA estimates of c-C$_4$F$_8$ emissions from U.S. the SC industry ([https://www.epa.gov/ghgreporting/data-sets](https://www.epa.gov/ghgreporting/data-sets), Subpart I) also only indicates a small increase from 0.005 Gg/yr (2012-2013) to 0.006 Gg/yr (2014-2015 and onward). We would rather not add this detail information to the revised manuscript. While we may not be able to explain all variability, the overall correlation between $c$-C$_4$F$_8$ and global HCFC-22 FS production is strong (R$^2$ = 0.97).**

*Other typographical/formatting comments:*

*For Figure 2, it would be useful to include the black diamonds in the legend. It also appears that the emissions data is offset on the x-axis compared to the production data, making it appear that there are two additional years of emission data compared to production data (but there is only one additional data point). It would be easier to read if they were not offset.*

**We thank the reviewer for noticing these two issues, which we have fixed in the revised figure and manuscript (the missing legend entry and the accidental ½ year shift of $c$-C$_4$F$_8$ emissions were fixed).**